# DNA Nanodevice-Based Drug Delivery Systems

**DOI:** 10.3390/biom11121855

**Published:** 2021-12-10

**Authors:** Chaoyang Guan, Xiaoli Zhu, Chang Feng

**Affiliations:** 1Center for Molecular Recognition and Biosensing, School of Life Sciences, Shanghai University, Shanghai 200444, China; 973861642@shu.edu.cn; 2Department of Clinical Laboratory Medicine, Shanghai Tenth People’s Hospital of Tongji University, Shanghai 200072, China

**Keywords:** drug delivery, nanodevice, DNA framework, stimuli response, controllable release

## Abstract

DNA, a natural biological material, has become an ideal choice for biomedical applications, mainly owing to its good biocompatibility, ease of synthesis, modifiability, and especially programmability. In recent years, with the deepening of the understanding of the physical and chemical properties of DNA and the continuous advancement of DNA synthesis and modification technology, the biomedical applications based on DNA materials have been upgraded to version 2.0: through elaborate design and fabrication of smart-responsive DNA nanodevices, they can respond to external or internal physical or chemical stimuli so as to smartly perform certain specific functions. For tumor treatment, this advancement provides a new way to solve the problems of precise targeting, controllable release, and controllable elimination of drugs to a certain extent. Here, we review the progress of related fields over the past decade, and provide prospects for possible future development directions.

## 1. Introduction

Cancer is regarded as one of the diseases challenged by modern medicine. Of note, although it is extremely ambitious to develop its specific drugs, we can adopt to optimize the pharmacokinetics and biodistribution of less specific drugs, especially the targeted delivery of tumors [1]. Therefore, stimulus-responsive drug delivery systems (DDS), which are exploited for smart medical treatment, have caused deeply concerned in twenty years. Due to the peculiarity of tumor microenvironment (TME), the concentration of most functional reagents delivered to tumor sites in the past few decades has been shown to be low. Some biological molecular drugs, such as antibodies, microRNA (microRNA), and small interfering RNA (siRNA), are easily degraded by enzymes or it is difficult to penetrate the cell membrane into the cells, greatly reducing their anticancer function; in addition, even adverse side effects and toxicity occurred [2,3,4,5,6]. By this token, the safety and accuracy of drug delivery are considerably critical for treating sophisticated and heterogeneous diseases such as cancer. Furthermore, an ideal drug carrier is consistently desired to defend the drug from degradation, to load one or more drugs, and to target specific delivery and release.

As a novel drug delivery system, DNA nanomaterials harbor tremendous potential in delivering multiple drugs to the ultimate target to implement combined therapy. Self-assembled DNA can be constructed into distinct geometric shapes, enter the tumor region, and pass through the cell membrane owning to its programmable structure and strict base complementary pairing principle. In addition, the DNA structure can also modify multiple functional ligands, such as targeting aptamers, gene sequences, imaging probes, etc., and can even be programmed to be capable of responding to external stimuli (such as light, heat, and magnetism) [7,8,9,10], or responding to the stimulus of pathological environment (such as pH value and enzyme concentration), being activated and releasing the carried drugs as needed, which solves the problem that the drug cannot be released after premature leakage or reaching tumor cells in the delivery process. Ultimately, DNA nanocarriers will be degraded and eliminated in vivo [11,12,13], reducing the potential risk of biotoxicity. In terms of clinical application, the latest evaluation results on drug delivery by DNA nanostructures indicate that the barriers to the development of DNA nanostructure-based drug delivery are likely to be primary technical, regulatory, and ethical rather than financial [14]. Therefore, solving the related technical problems is given priority.

For decades, researchers have designed and constructed various DNA or DNA-based nanomaterial structures by virtue of using the intrinsic properties of DNA itself [15,16,17,18,19,20,21,22,23,24,25,26,27]. Compared with other nanomaterials, due to its good controllability, biocompatibility, easy modification, and programmability, the drug delivery efficiency of DNA nanomaterials in the tumor field undoubtedly provides a new prospect for constructing nano drug carriers from bottom to top [28,29,30,31,32,33]. In short, from targeted drug delivery, cell uptake to stimulus-responsive release, a large number of studies have confirmed that DNA nanomaterials can effectively load drugs with the help of some functional elements, and accomplish efficient delivery. However, due to the discrepancy in structure and function, distinct types of stimulus-responsive nanomaterials harbor different application fields and effects. This review is based on the development of DNA nanodevices and, as shown in Figure 1 in recent years, lists several kinds of DNA nanostructures used for drug carriers, and summarizes the applications of DNA nanodevices responding to physical, chemical, and biological stimuli in the fields of tumor treatment and biological imaging, and looks forward to the future development, with a purpose towards the vigorous development of DNA nanotechnology involving multi-disciplinary fields and its early application in clinical practice.

## 2. DNA Nanostructures for Drug Carriers

Initially inspired by the upward and downward sliding of the Holliday intermediate along a pair of homologous DNA chains, about 30 years after the mysteries of the DNA double helix were solved, Nadrian C. Seeman [34] initiated the historical development of DNA nanostructures based on the principle of base complementary pairing. They exploited building blocks to construct tile and design connections for loading some biological macromolecules. In 2006, after the founder Rothemund [35] developed the landmark DNA origami, scientists studied DNA origami mainly with the desire that it could be used as a raw material to build nano-models, rather than a carrier of genetic information. How to take advantage of DNA self-assembly nature to design and construct more complex DNA-based 2D or 3D nanostructures has attracted substantial interest from researchers. From 2D smiley faces to 3D geometric objects and letter building blocks, origami technology is becoming more and more advanced. Over the past few decades, resulting from the emerging medical technology requiring personalized treatment with accurate diagnostic data, the demand for molecules that have a high degree of specificity and affinity for their target molecules has been increasing [36]. The emerging DNA nanobots [37,38,39] have achieved dynamic mechanical functions on the basis of DNA origami technology, which can carry drugs along the designed path to accurately reach the position of the lesion for accurate drug administration, providing a potential application prospect for the adjustable loading and release of drugs.

In this case, DNA can simultaneously load multiple drugs with the characteristics of nanoscale size and base complementary pairing, as well as load cargo of distinct sizes and types according to different circumstances. The high addressability of DNA nanostructures enables accurate control of the valence state and position of the loaded molecules. Functional DNA shows nanostructures consisting of three-dimensional complex structures, including tetrahedral, cubic, spherical, and polyhedral structures. However, in addition to being used as a biomaterial alone, it can be combined with or cooperated with other nanomaterials such as metal nanoparticles, nanorods, and get access to maximize therapeutic effects.

### 2.1. DNA Framework

#### 2.1.1. DNA Origami Structure

DNA origami refers to the binding and spreading of a long single strand of DNA into a certain shape like folding a piece of paper, which may be known as “DNA rope folding”. Rothemund [35] made the single-stranded DNA of M13mp18 bacteriophage as the nucleating chain, known as the scaffold chain or skeleton chain, and the short chain used for binding was also known as the staple chain or auxiliary chain. DNA origami establishes the pattern by folding the entire chain back and forth in a raster-filled fashion so that it is full of the designed pattern. The average molecule is carried on a “staple” because of each DNA nanostructure containing about 200 staples and can accurately carry the “cargo”. The method greatly enlarges that structural functionality of DNA, and has a high assembly success rate and yield. Nonetheless, each design requires the synthesis of hundreds of different DNA chains, which intensively expands the complexity and cost of the experiment and limits the large-scale production and application.

Doxorubicin (Dox) is a regularly operational anti-tumor drug to treat various tumors by inhibiting DNA synthesis. Due to the structural specificity of inserting DNA double-stranded G-C base pair, Dox is usually used to test the delivery efficiency of drugs carried by DNA nanocarriers. Although Dox is one of the most effective chemotherapy drugs approved by FDA, it can effectively kill cancer cells while damaging normal cells, and its side effects and low selectivity have resulted in dose-limited Dox-based treatment. For instance, Dox was non-covalently attached to DNA origami nanostructures through intercalation [40], and Ding and his colleagues present a novel drug carrier system based on self-assembled, spatially addressable DNA origami nanostructures. The results showed that the DNA origami/Dox complex not only had significant cytotoxicity to normal human breast cancer cells (MCF7), but more importantly, had significant cytotoxicity to Dox-resistant MCF7 cells, resulting from the increased uptake of Dox by cells using origami vectors and the redistribution of Dox at the action site. In a follow-up project, Ding and his colleagues studied the therapeutic effects of the origami/Dox system in vivo, particularly in multidrug-resistant (MDR) cancer cells. Hogberg’s research group [41] tested two DNA origami nanostructures with different degrees of distortion on three different breast cancer cell lines (MDA-MB-231, MDA-MB-468, and MCF-7) to regulate the drug encapsulation efficiency and release rate. This demonstrated the feasibility of controlling drug loading and release by modulating the DNA origami nanostructures, providing a new avenue for obtaining more tailored pharmacokinetic DNA nanocarriers. Wu et al. [42] reported a DNA origami nanostructure (DON) modified with targeted ligands (Figure 2a), which could be used as an antibody-drug conjugate (ADC)-like carrier for targeted treatment of prostate cancer. Dox was loaded onto DON by embedding dsDNA. Liu et al. [43,44] built up a series of DNA nanostructures as a co-delivery vector of RNA interference (RNAi)/p53 and chemodrugs for combined therapy, to combat multidrug-resistant tumor (MCF-7R) in vitro and in vivo without apparent systemic toxicity. In addition, the carrier DNA structure also provides a large number of dense insertion sites for the low-molecular mouse carbazole derivative BMEPC in tumor cells.

In addition to Dox loading, CpGs (unmethylated cytosine-phosphate-guanine dinucleotides) located in the microbial genome can be used as a safe and effective adjuvant for immunotherapeutic vaccines. Nucleic acid sequences with unique biomedical functions like CpG also include antisense nucleic acids, siRNA, and miRNA, which can be designed as part of DNA nano-devices [45]. Bacterial DNA containing CpG motifs can be identified by the innate immunity of mammals and trigger strong immune responses [46,47]. However, single CpG dinucleotides harbor poor uptake ability by cells and are easily degraded by nucleases [48]. Scientists have formulated a variety of schemes to solve this problem. The Liedl team [49] tested the immune response induced by 30 spiral tubular DNA origami loaded with 62 CpG sequences in spleen cells. CpG DON was anchored on the surface of designed hollow DNA nanotubes. They found that the origami tube modified with CpG could trigger higher immune stimulation than a standard carrier system (such as liposome). The complex was shown to be localized to the endosome and to promote cytokine production with no detectable toxicity or impact on splenic cell activity. This suggested that the stability and compactness of the DNA nanotube skeleton effectively resisted degradation of CpG DON by nucleases. Furthermore, Castro et al. [50] also assembled a rod-shaped DNA origami carrier controllably loaded with daunorubicin drugs, which also belong to anthracyclines, for resisting a certain dose of daunorubicin resistance in leukemia cell models. This robust drug delivery method can increase drug retention in cells and reduce drug resistance of leukemia cells, providing a theoretical basis for the design and transformation of DNA origami to examine other drug delivery systems for clinical blood circulation tumors.

Triangular DNA nanostructures offer unique advantages over square and tubular structures with reasonable geometry and a size of about 120 nm on each side. In the work of Pei et al. [51], triangular DNA origami was also exploited as the targeted DDS for cancer treatment, showing favorable biocompatibility and stability in cell culture medium within 24 h. In addition, the DNA origami structure conjugated with the multivalent aptamer can efficiently deliver the anticancer drug Dox to targeted cancer cells because of its targeting function, and the system also illustrates a prominent therapeutic effect in vitro. In order to further develop the targeting efficiency of DNA origami nanostructures, folic acid functionalized triangular DNA origami can also effectively target Dox to breast cancer cells overexpressed by folate receptor (FOLR1) [52], which selectively ingests folic acid-modified DNA nanostructures through receptor-mediated endocytosis, and Dox is released into the nucleus to induce cytotoxicity and cell death. Ding research group [53] not only proved that the triangular origami structure showed the optimal passive targeted accumulation of tumors, but also DNA origami containing Dox showed significant anti-tumor efficacy and no side effects in nude mice with breast tumors labeled with green fluorescent protein in situ. It has been proved that functionalized DNA origami, as an excellent biocompatible and targeted drug delivery carrier, has great potential in the treatment of cancer. However, there is still a lack of relevant research to utilize the unique potential of DON to control drug space tissue, which may be very important for optimizing the efficacy and enhancing the safety of drugs [54].

Nanobot is one of the largest applications of DNA origami technology. At present, nanobots have revealed great potential in drug delivery and disease treatment. In 2018, Yan team and researchers from Chinese National Center for Nanotechnology jointly developed a typical DNA nanorobot delivery system based on DNA origami technology [37]. The outside of the nanorobot is a DNA aptamer combined with nucleolin (nucleolin is mainly expressed on the surface of tumor cells), and the lumen is loaded with thrombin that can lead to thrombosis and kill the tumor (Figure 2b). Externally, via recognizing the tumor microenvironment signals, the nano-robot can accurately deliver the drugs to the blood vessels near the tumor, so that the drugs can be accurately released to form thrombus near the tumor and block the supply of the tumor, leading to tumor necrosis and inhibition of tumor growth. This year, Nie Zhou’s team [55] reported for the first time that DNA nanobots, which can walk on the interface of mobile cell membrane and have the function of driving cells, could regulate cell rearrangement by constantly activating receptor signaling pathways and super-sensitively control cell migration behavior. Additionally, in this design, by customizing different DNA enzyme-based driving modules, the conditions of different selective environments can be controlled for the operation of DNA robots, and then the cell behaviors are controlled orthogonally, which further verifies the multi-power of this design strategy. In conclusion, this research provides a new strategy to transform nano-scale molecular manipulation of DNA robots into micro-scale behavior of living cells, which is extremely hopeful to promote the development of cell-based nano-scale precision medicine in the future.

#### 2.1.2. DNA Tetrahedral Nanostructures

DNA tetrahedral nanostructure (TDN) is the simplest DNA cube synthesized by Turberfield et al. [56] in 2005, and consists of four DNA oligonucleotides. This three-dimensional nanostructure has high mechanical rigidity, nuclease degradation resistance and no side effects, and the cytoplasmic membrane is permeable to its typical spatial structure, making it a recognized candidate for a drug carrier. Studies have shown that Dox carried by DNA tetrahedrons can effectively avoid cellular efflux and reverse the drug tolerance of tumor cells. This simple and efficient DNA regular tetrahedron drug delivery system is considered to be able to improve the relatively complex construction of the existing DNA origami drug delivery system [57].

In 2016, Kim and Lee et al. [58] developed a self-assembled mirror DNA tetrahedral structure, which greatly enhanced the stability of DNA vectors. l-DNA is a mirror image of natural d-DNA, and both have the same thermodynamic properties. However, mirror image DNA can solve the problem of poor serum stability of natural d-DNA, which has symbolic effects on pharmacokinetics and biological distribution. Their research illustrates that the mirror DNA nanostructures can selectively deliver anti-cancer drugs to specific tumor cells and tissues to enhance the permeability of tumor cells and tissues, displaying a stronger anti-cancer result than the traditional methods. A year later, their research group [59] explored the in vivo and in vitro behavior of DNA tetrahedrons as drug carriers (Figure 3a). The laser scanning confocal microscope (LSCM) was used to observe the change of cell uptake over time in vitro, and pharmacokinetic studies were conducted in vivo. It was found that, compared with free Dox, the clearance rate of Dox loaded on DNA tetrahedron modified by folic acid was decreased, and the initial concentration, blood circulation half-life, and circulation time were increased. This low dose administration via DNA nanostructures in particular enhances the accumulation of drug at the tumor and reduces cardiotoxicity.

In 2011, Fan et al. [60] developed a polyvalent DNA tetrahedron with unmethylated CpG motifs added, demonstrating that this DNA nanostructure complex was resistant to nuclease degradation and could maintain its intact structure in fetal bovine serum and cells for at least several hours. Notably, the complex was also able to efficiently and non-invasively enter macrophage-like cell RAW264.7 in the absence of transfection reagent. After cell uptake, TLR9 recognized CpG motifs and then activated the downstream pathway to induce immune stimulation [61], resulting in high-level secretion of various pro-inflammatory factors such as TNF-α, IL-6, and L-12. In addition, the more CpG on the DNA tetrahedral structure, the stronger the stimulation activity. Furthermore, at low concentrations of TDNS (usually <250 nM), various living cells (such as RAW264.7 cells and L929 fibroblast-like cells) showed no obvious cytotoxicity or adverse reactions [62,63,64], and even higher concentrations promoted cell proliferation or migration activities, further proving that TDNs had favorable biocompatibility [65,66].

The combined application of TDNs and various other small molecules have been elaborated by multiple research institutes in recent years, which solves the key problems in the development of small molecule drugs, such as poor membrane permeability, no targeting function, and instability in physiological environments. Charoenphole et al. [67] introduced AS1411 aptamers (AS1411 nucleic acid aptamers capable of specifically binding to nucleolin) into DNA tetrahedral nanostructures to enhance cell uptake and specifically inhibit cancer cell growth in the absence of transfection reagents (Figure 3b). The Lin team [68] presented the multifunctional TDN-based delivery system that can transport antisense PNA (asPNA) in their overview, DNA aptamers, and small-molecular-weight drugs into bacteria or targeted living cells. In their scheme, TDN-based delivery systems are established via four different modification approaches: using antisense peptide nucleic acid instead of single-stranded short sequence; attaching an aptamer at the tip; directly incubating with small molecular drugs; and a coating protective agent. Undoubtedly, these TDNs modified complexes have broad prospects in targeted therapy, anticancer and antibacterial activities, and promotion of uptake.

In addition, siRNA (small interfering RNA) can be investigated to treat RNA interference (RNAi) in eukaryotic cells by targeting the induction of complementary mRNA degradation. Anderson et al. [13] showed that self-assembled DNA tetrahedral nanoparticles with a well-defined size can deliver siRNAs into cells and silence target genes in tumors. They demonstrated that gene silencing occurs only when the ligands are assembled with proper spatial localization, and superlative delivery occurs when the three folic acid molecules are modified tetrahedrally to maximize local density. In addition, in vivo tests showed that these nanoparticles exhibited longer blood circulation time, mainly in tumors and kidneys. This indicates that DNA tetrahedrons, assisted by functional elements, are expected to become universal platforms for siRNA delivery. Among the strategies proposed by Chen et al. [69] for cancer treatment using tetrahedral DNA cage-loaded ruthenium polypyridyl complexes (RUPOP), the binding of biotin enables the system to enhance specific cellular uptake, drug retention, and cytotoxicity against HepG2 cells. The RuPOP-DNA cage was internalized and then entered the nucleus, the DNase response triggered the release of drugs and induced ROS-mediated apoptosis. In the nude mouse model, the system specifically gathered at the tumor site and effectively inhibited tumor growth.

Cisplatin and platinum-based drugs are widely used in clinical practice, and the antitumor mechanism is mainly the interaction with DNA base pairs. However, patients need to endure a series of adverse reactions such as nephrotoxicity, neurotoxicity, and ototoxicity during chemotherapy, and the delivery of platinum drugs may be conducive to reducing drug toxicity and improving the treatment effect. The Ding team [70] embedded platinum-based drug 56MESS into a double-stranded DNA tetrahedron by the intercalation method, and coupled with a nano antibody targeting the inhibition of epidermal growth factor receptor (EGFR, one of tumor markers), in order to implement multi-drug combination therapy. We found that the DNA nanoplatform combined with the nanobodies not only showed outstanding selectivity for tumor treatment, but also showed no palpable cytotoxicity. It is well known that tumor cells have tumor antigens, and the immune system of the body can distinguish tumor cells from normal cells by recognizing tumor antigens. Tumor vaccines inhibit tumor development by expressing specific and immunogenic tumor antigens (such as polypeptides, nucleic acids, and so on) and activating the body’s anti-tumor immunity with the assistance of adjuvants. In this case of co-delivery based on DNA nanostructure platform, Chang et al. [71] also assembled the model antigen (streptavidin molecule STV) and CpG adjuvant (mouse-specific ODN-1826) into a nano-scale complex, which could generate a strong, specific, and durable immune response against the antigen without affecting the DNA nano-scaffold. As shown in Figure 4, Setyawati et al. [72] have produced stoichiometric DNA tetrahedral nanostructures that can be utilized to co-deliver Dox and specific EGFR-targeting cetuximab. The apices of each etrahedron can carry 1–3 cetuximab. As the number of carried antibodies increases, its lethality to tumor cells will also increase. When carrying three antibodies, the other vertex is used to bind to the fluorescent probe, but in theory a fourth antibody can also be conjugated. This synergistic effect of dual immunotherapy once again demonstrates the infinite potential of DNA nanostructures for the safe delivery and maximization of drug efficacy, as well as the construction of a common platform for a variety of vaccines. In the delivery treatment of natural protein, Li et al. [73] designed a fully enclosed divalent aptamer tetrahedral DNA framework (asdTDF) delivery system, in which a protein drug was reversibly wrapped in the framework cavity, and the auxiliary blocking of ligase enabled TDF to resist nuclease degradation, thus enhancing its stability. The participation of bivalent aptamers enables them to have targeting and efficient transmission ability, and the natural therapeutic protein will undergo trigger release followed by the endogenous glutathione cutting the chemical bond.

It is not difficult to see that the preparation process of the DNA tetrahedron is simple, the yield is high, the size can be dynamically adjusted, and the DNA tetrahedron has good biocompatibility and penetrability so as to stand out in the manufacture of nano devices. Although DNA tetrahedral nanostructures do represent an ideal and simple three-dimensional DNA nanomaterial, its stability in vivo remains to be strengthened.

#### 2.1.3. Other DNA Framework Structures

There are a variety of DNA nano-frame structures that can be investigated for drug delivery. In addition to the several structures above, the most famous one is that Ma et al. [74] designed a DNA icosahedron responsive to telomerase, which is composed of two pyramidal DNA cages connected with telomerase primers and telomere repeat sequences, with platinum nano-drugs encapsulated in the DNA structure. The system can respond to telomerase in tumors and perform accurate release to enhance the anticancer effect while reducing toxicity for normal tissues.

In 2012, the Chruch team developed a logic gate-controlled DNA nanorobot [75] for targeted transport of molecular payloads, including antibodies. Complex-shaped structures formed by manipulation of long DNA chains by binding to shorter “backbone” chains that can transmit payloads, such as gold nanoparticles or fluorescently labeled antibody fragments. They used origami technology to design a hexagonal DNA barrel that could respond to various switches via aptamer-encoded logic gates. These nanobots are designed to open in response to specific cell surface proteins, releasing molecules that trigger cell signals. In 2014, Sleiman’s team produced a 3D DNA prism integrating gene silencing of antisense oligonucleotide chain (ASO) at locuses 1, 2, 4, and 6 [76]. Specifically, the addition of antisense strand makes it easier to induce gene silence and maintain gene knockout level in mammalian cells, which we know is attributed to that intensive stability of DNA nanostructure. Later, the research group further designed, synthesized, and optimized DNA “nanosuitcases” [77], which can encapsulate siRNA and release it after recognizing mRNA or microRNA, demonstrating the potential of dual or synergistic treatment. They found that the DNA framework protects cargo from site-specific cleavage and nuclease degradation, releasing cargo on demand while maintaining biological conditions.

Previously, Huang’s research group [78] created MUC1-aptamer-DNA icosahedral using a five-or six-point star-like motif connected by sticky ends. The G-C base pair of Dox embedded in the icosahedral can be controlled released by environmental pH, and after incubating with MUC1 positive cells, the intracellular internalization rate and cytotoxicity can be significantly improved. Lately, Tan [79] has adopted a layered self-assembly strategy and created a core-shell nanostructure consisting of a DNA octahedral framework and CA4-functionalized Sgc8c-aptamer (CA4-FS). CA4-FS can be loaded onto each side of the DNA frame, and a different number of CA4-FS can be accurately loaded onto the DNA octahedron by reasonable design (Figure 5a). The results of the comparative study showed that CA4-Oct had sufficient biological safety and was the most effective anti-cancer therapy (Figure 5b). This indicates that the strong permeability and retention of DNA nanostructures, together with the active targeting of Sgc8c aptamers, allow highly selective chemical drug delivery and effectiveness in vivo imaging and therapy.

### 2.2. Spherical Nucleic Acid

Spherical nucleic acid (SNA) is a new nanomaterial composed of oligonucleotide chains which can be densely packed and highly oriented in nanoparticle templates, with a head group attached to the nanonucleus and a tail group extending into solution, and the general size is 10–50 nm. In the chemical synthesis process, gold (Au), silver (Ag), Fe_3_O_4_, quantum dots, platinum, SiO_2_, and liposomes were adopted as the nanoparticle nuclei, and the dispersed nucleic acids included DNA, siRNA, micro RNA, peptide nucleic acid (PNA), locked nucleic acid (LNA), etc. Since AuNPs can be easily synthesized in different particle size ranges with high extinction coefficients, they can be functionalized with a variety of chemical reagents and display a clear catalytic performance, they are often viewed as core materials [80]. It is well known that SNA can resist the degradation of nucleases with good stability and strong binding force, and can be used for intracellular probe, drug delivery, gene regulation, molecular diagnosis, intracellular imaging, tumor treatment, and the like.

Despite siRNA being a promising drug in various diseases, there are still some challenges in its delivery into cells. Owing to the fact that SNA can naturally and rapidly enter more than 50 cells, the use of SNA as a single entity transfection agent to modify the siRNA double strand has been used for the treatment of numerous diseases, yet the mechanism leading to gene silencing is still ambiguous. In general, gene silencing refers to the regulation of the expression of relatively large genes through a small segment of nucleic acid molecules, such as siRNA, ribozymes, DNA enzymes, antisense oligonucleotides (ASO), CpG, and aptamers. However, we can block the gene expression that causes cancer occurrence through gene silencing of “spherical nucleic acids” to fulfill the purpose of treating tumors. In light of the mutated gene, dozens of small RNA molecules are designed. If these small RNA molecules are sent to tumor cells, the cancer-causing genes will be degraded and the normal genes of the human body will not be affected. This achieves the purpose of precise tumor treatment, from this perspective, such drugs can generally treat major tumors. Ruan et al. [81] formulated an excellent biocompatible spherical nucleic acid conjugate by grafting siRNA onto the surface of a nucleus consisting of spherical DNA nanostructure (DNA nanoclew, DC). After being ingested by cancer cells, the SNA nanoparticles release siRNAs through Dicer enzyme cleavage. In vitro experiments have proved that the SNA exhibits potent gene knockout at both mRNA and protein levels with negligible cytotoxicity. Yamankurt et al. [82] conducted research on the cytoplasmic process of siRNA-SNAs and how they lead to gene knockout. Dicer cleaves the modified siRNA duplex from the surface of the nanoparticle, and the liberated siRNA subsequently functions in a way that is dependent on the canonical RNA interference mechanism. Based on this mechanism, the researchers designed a structure with siRNA density one order of magnitude higher, and the results showed that the increase in nucleic acid content can reduce the cytotoxicity of SNA without reducing the bioactivity of siRNA. Interestingly, the recent work of Deng et al. [83] has turned SNA synthesis into a simple task to fulfill the instantaneous (in seconds) synthesis of SNAs with record high DNA density, enabling us to explore the physical, chemical, and biological effects of SNA with ultra-high DNA density in the future. Previous studies have found that the stability of DNA-AuNPs is highly dependent on the density of modified DNA, by virtue of the densely arranged AuNPs having higher local salt concentration than the loosely arranged AuNPs, and high-density DNA-AuNPs also perform better gene silencing than low-density DNA-AuNPs [36].

Glioblastoma (GBM) is a typical primary malignant brain tumor. The existence of the blood–brain barrier (BBB) is vital to restrain harmful substances from entering the brain from the blood. However, the BBB also prevents the transfer of most small molecule drugs and macromolecules, severely limiting the treatment of central nervous system (CNS) diseases. Therefore, it has been difficult to prescribe a drug that can break the blood–brain barrier to treat glioblastoma in the past decade. In 2017, Sita et al. [84] developed the nanoconjugate SNA for blood–brain barrier/blood tumor barrier penetration to deliver siRNA and miRNA to intracranial glioblastoma tumor sites. Their model O6-methylguard-DNA-methyltransfer (MGMT) can vertically and non-invasively measure the response of MGMT gene knockout to MGMT-specific SNA treatment in vivo, and identify the most resultful siMGMT-SNA protocol by optimizing the SNA carrying MGMT siRNA specificity in animals, as monotherapy and in combination with temozolomide (TMZ). Analysis of the biological distribution and pharmacokinetics of the SNA showed that rapid absorption and significant retention within the tumor increased the antitumor activity of coadministered TMZ. In addition, it is very noteworthy that Kumthekar et al. [85] of Northwestern University in the United States recently developed an SNA drug (NU-0129) consisting of siRNA arranged around gold nanoparticles (AuNP). These siRNAs specifically target the mRNA of Bcl-2L12 genes and reduce the expression of Bcl-2L12 gene by infecting RNA. The research group has been studying spherical nucleic acid drugs since 1996 and has conducted toxicological and toxicokinetic studies in non-human primates. Since the implementation of the Phase 0 clinical trial in 2017 (the Phase 0 clinical trial is the first trial in which drugs are applied to a very marginal number of subjects after animal trials), the eight patients enrolled in the trial were given a small dose of NU-0129 intravenously one day before surgery. After the surgery, scientists observed whether the NU-0129 drug passed the blood–brain barrier and was swallowed by tumor cells. Clinical trials have demonstrated that NU-0129 penetrates the blood–brain barrier, inhibits oncogene expression levels, and leads to tumor cell apoptosis or programmed cell death, even at small doses. The results of this research are very meaningful, and it is the first time that nanotherapies penetrate the blood–brain barrier through intravenous infusion and change the genetic mechanism of tumors to cause the death of tumor cells. Although this novel SNA drug platform is currently only an early clinical trial, it represents the delivery of a revolutionary new drug, and it is expected that the drug will soon carry out a phase I clinical trial, which is expected to be applied to nervous system diseases such as Alzheimer’s disease, Parkinson’s disease, and Huntington’s disease in the future. We may compare this drug to another “PD-1”, which can be made into many drugs against many cancer-related genes. Thus, for complicated tumor gene mutations, we may consider achieving a better controlled delivery effect by using a drug combination.

From the above, we have found that normal linear nucleic acids generally cannot enter cells or cross the blood–brain barrier, but this spherical nucleic acid can achieve the above process. The key to the success of SNA drugs is that it has a nano-3D spherical structure and the density of nucleic acids. RNA chains are tightly adsorbed and surrounded like a protective shell on gold nanoparticles with a diameter of only 13 nm. Nucleic acids can be densely wrapped and form micro-spherical structures. In view of the fact that some anticancer drugs may cross the blood–brain barrier and have significant side effects on the central nervous system, the spherical nucleic acid system described by Bousmail [86] in another article is used to deliver the anticancer drug BKM120 for the treatment of chronic lymphocytic leukemia (CLL), with minimal leakage through the blood–brain barrier. The BKM120-loaded DNA nanoparticles [86] can promote the apoptosis of lymphocytes in patients with primary CLL and serve as sensitizers for other anti-tumor drugs (such as Dox) without causing non-specific inflammation, and have a long circulation time (up to 24 h), systemic distribution, and accumulation at the tumor site. The Li team [87] has recently reported a DNA biosensor for the detection of flap endonuclease 1 (FEN1, one of the biomarkers of tumors) and in vivo and in vitro imaging. This DNA nanostructure has good stability, sensitivity, and specificity for FEN1 detection, and is expected to be used as a precise therapeutic agent for space-time controlled targeted drug administration. The oligonucleotides designed on the spherical nucleic acids can encode the substrate chain of FEN1 for fluorescence sensing, the aptamer sequence of AS1411 for cancer cell recognition and targeting, and the GC-rich sequence for loading the anticancer drug Dox. Thus, the fabricated spherical nucleic acid can accurately target cancer cells and tumors and explicitly show FEN1 levels for early tumor diagnosis.

In addition to taking AuNP as the core, the research group of Ke Zhang from Northeastern University [88] reported a self-assembled spherical nucleic acid vector capable of reducing skin pigmentation, which demonstrated enhanced ability to penetrate the epidermis and dermis after skin administration. The carrier contains a double-functional oligonucleotide amphiphilic chain consisting of an antisense oligonucleotide targeting MC1R and a tyrosinase inhibitor prodrug, forming a spherical micelle and a compact DNA shell (Figure 6a), wherein the two synergists promote oligonucleotide cell absorption, improve drug solubility, and promote skin penetration and the like. The granule [88] has the capacity to decrease the melanin content in B16F10 melanoma cells and has an effective anti-melanin production effect in a mouse model of hyperpigmentation induced by ultraviolet B. This group also previously reported an SNA [89] using drugs as the core, which utilizes amphiphilic DNA- anticancer drug paclitaxel (PTX) to self-assemble into micelle nanoparticles with a structure similar to SNAs in solution by covalent attachment. The nucleic acid component in the SNA is not only used as a delivery carrier for drug components, but also as a payload for intracellular gene regulation and treatment. The nucleic acid component enters cells 100 times faster than free DNA, and has enhanced nuclease resistance. Compared with free drugs, it has little discrepancy in cytotoxicity. The results undoubtedly suggest that people can only use the payload itself to achieve gene and drug targeting without necessarily requiring other carrier systems. Differing from the drug core, Sinegra et al. [90] synthesized lipid-core lipid nanoparticles SNA(LNP-SNA) for the delivery of DNA and RNA to targeted sites in the cytoplasm. Its surface DNA accumulation, G sequence enrichment, and core lipid content increase can all lead to the enhancement of LNP-SNA activity. The optimized LNP-SNA can reduce the siRNA concentration required for silencing mRNA by two orders of magnitude. Their experiment found that the LNP-SNA structure was different from the conventional liposome SNA in efficacy and biological distribution (Figure 6b) and therefore could be used to target tissues.

SNAs typically embody an emerging nucleic acid structure that can enter cells in large amounts without the need for transfection reagents, and even cross the epidermis, BBB. In recent years, SNAs has been used as a multifunctional material, which has great potential in molecular diagnosis and gene therapy by combining with various small molecules and drugs. It is possible that oligonucleotide-based therapy can be customized for the diagnosis and treatment of many diseases, and in the future, SNAs may be used simultaneously as an individualized on-demand therapy for different diseases. At the same time, considering the special environment in vivo, the manufacture of biocompatible SNA nanodevices to maximize the efficacy in vivo still needs to be resolved urgently.

### 2.3. DNA and Other Nanomaterial Complex

So far, we have introduced the unique structure based on DNA, and these structures show unique biological and chemical properties; it is not difficult to see that scientists have realized a high degree of programming of DNA structure design. While functional inorganic nanomaterial such as gold, quantum dots and fluorescent nanoparticle have been studied for various forms of hybridization with DNA origami in order to exploit that beneficial property of DNA to explore applications in the biomedical field, including biosensors and drug delivery vehicles. Of note, when these functional DNA are combined with nanoparticles, they maintain their structure in an extremely stable manner, which allows DNA-conjugated nanoparticles to be applied as potential materials for biological applications.

Gold nanoparticles have good biocompatibility, stability, special size structure effect, and local surface plasmon resonance (LSPR) characteristics, and are used frequently to build a variety of nano-devices. DNA-AuNPs not only possess the optical and physicochemical properties of AuNP, but have the self-assembly ability and programmability of DNA. Some of the DNA structures mentioned in the above section, especially SNAs, are products of the covalent conjugation of DNA to AuNP. Moreover, fluorescent dyes ordinarily occupy an irreplaceable position in cell analysis. With the maturity of DNA synthesis technology, it has been considered to modify various fluorescent dyes on DNA chains for subsequent use of fluorescence microscopy to track the circulation, distribution, and lifetime of DNA nanodevices in living cells. Typically, fluorescent nanoparticles are combined with other functional molecules to bind to DNA nanostructures for simultaneous imaging and delivery. Tan et al. [91] created aptamerally linked fluorescence resonance energy transfer (FRET) nanoflowers (NF) based on rolling ring amplification for multicellular imaging and traceable targeted drug delivery. Here, DNA can be covalently combined with three FRET fluorescent dyes to facilitate NF that can emit different fluorescence emissions with uniform performance, high fluorescence intensity, and high photostability. Differing from the traditional assembly method, the NF assembly does not depend on the template sequence after using the rolling circle amplification (RCA) method. RCA is an isothermal enzymatic reaction that can rapidly synthesize DNA, avoiding the original complex design of DNA building blocks that are assembled into nanostructures through base pairing [91].

Zhu et al. designed a double-targeted and cascade-enhanced intelligent response nucleic acid therapy for Bcl-2 gene silencing [92], using graphene oxide (GO) as the nanocarrier to load an elaborately designed DNAzyme that can target and silence Bcl-2 mRNA. When DNAzyme binds to Bcl-2mRNA, the enzyme activity activates and cleaves the substrate oligonucleotide to produce anti-nucleolin aptamer AS1411. With the help of cofactor K^+^, AS1411 is allowed to form a stable dimer, G-quadruplex, to bind to and inactivate nucleolin, and inactivate Bcl-2mRNA via cascade reaction. The results demonstrated that, even without any chemotherapy drugs, after the nucleic acid drugs were administered, the apoptosis of tumor cells could be observed at the cell level, and a visible shrink of the tumor in vivo could be observed without robust side effects [92]. Therefore, the novel nano composite material is feasible to be applied as an intelligent response therapy tool for synergistic treatment and imaging of various cancers in the future.

The above case studies in this chapter demonstrate the immense potential of the combination of programmable functional DNA with functional nanoparticles as a combination therapeutic agent. With the in-depth research on assembling DNA nanostructures, tremendous hopes are placed on the further development of this technology and biomedical applications.

## 3. Intelligent-Responsive DNA Nanodevice

Inspired by the structure and function of natural macromolecules with stimulation response characteristics in biological systems, people have attempted to design, build, and operate similar nano systems or devices at the molecular level to enable them to respond in time to complete the task under subtle environmental changes. Among them, DNA molecules play an important role from sensing to activation to response due to their unique programmability, predictability, and biocompatibility. In recent years, functional DNA molecules with favorable biological recognition ability have been used to design intelligent medical nano-devices, in order to expect to complete specific high-precision tasks in vivo, such as biosensing and imaging, molecular information calculation, and controllable drug transport. Intelligent responsive DNA nano delivery system also arises at the historic moment. By designing nucleic acid nanocarriers that release drugs in response to specific stimulation factors, researchers can solve the problem that drugs cannot be released after premature leakage or reaching tumor cells during the delivery process. Compared with the traditional drug administration, drug stimulation response and controlled release have the active targeting feature, which is conducive to improving the efficacy of the drug and reducing the toxic and side effects, and has great significance in elevating clinical medication.

At present, the receptor targets recognized by most nano-devices are not exclusive to disease tissues. The nano-devices mentioned above can generally target a variety of tumor diseases and do not have time controllability in the drug delivery process, thus weakening the specificity and accuracy of diagnostic and therapeutic applications. Therefore, how to regulate the biological recognition function of nano-devices in time and space to proceed with more accurate biological applications has attracted much attention. In the following sections, we summarize the development of intelligent responsive DNA nanodevices in recent years from physical, chemistry, and biology according to different types of stimulation factors, in order to provide a reference basis for more sophisticated device design in the future. The stimulation factors of the intelligent response-type DNA nano-drug delivery system studied in depth mainly include acidic pH in the tumor region or lysosome, near-infrared light, temperature, specific enzymes, and intracellular glutathione [36].

### 3.1. Physically Responsive DNA Nanodevices for Drug Carriers

Light stimulation does not need to meditate the shifts of complex physiological environment in vivo. It is a clean, non-invasive, and efficient type of external stimulation. The photoresponse wavelengths of DNA were mainly in the ultraviolet/visible (UV/Vis) and near-infrared (NIR) regions. The ultraviolet light was controlled to release mainly through photolysis, chain scission, and pH change, while the near-infrared light was converted by the photothermal method to release controllably. In addition, photodynamic therapy (PDT), a clinical treatment, mainly relies on the production of cytotoxic reactive oxygen species (ROS) under the irradiation of photosensitizers (PSs) [93]. ROS can oxidize lipids, amino acids, and protein, cause irreversible damage to cell membranes and important organelles, and induce cell death through apoptosis, necrosis, or autophagy [94].

Usually for the photolysis denaturation system, the vast majority of studies adopt an azobenzene molecule with a light-sensitive bond wedged into the DNA chain. When irradiating by UV light (300–380 nm) and cis-visible light, that azobenzene group (and its derivatives) can undergo a trans-to cis-reversible molecular transformation result in light-mediated controlled drug release. For example, Tan et al. [95] used photoactive azobenzene as a crosslinking agent in combination with DNA chains to construct a hydrogel that can store and release molecules and nanoparticles based on the rheological changes of the hydrogel, including dissolution of the hydrogel under UV irradiation to release drugs. Zhang et al. [96] grafted one end of DNA combined with an azobenzene group onto the surface of mesoporous silica nanoparticles (MSN) and modified the other end with AuNP. Local melting and hybridization studies using FRET under alternating UV/Vis illumination controlled the blocking and opening of the orifices by AuNP to control release. When the photosensitizer was inserted into DNA, it could lead to photo-chain scission. When Qu et al. [97] performed light irradiation on the constructed novel light-controlled carrier, the photosensitizer would generate ROS, leading to the end-capping of G-quadruplex DNA to be cleaved, thereby releasing the goods. This undoubtedly laid a foundation for the development of drug carrier systems for synergistic association treatment of chemotherapy and PDT, especially for cancer treatment with space/time control [97]. Photo-induced pH shift refers to the pH shift caused by the reaction of photosensitive substances under the light condition, thus transforming the structure of DNA. Zhao et al. [98] fixed the photosensitive substance malachite green (MGCB) at the nanochannels of MSN, and the i-motif DNA structure was grafted onto the surface of MSN to seal the orifice; i-motif was a special DNA four-chain structure, and only when two cytosine-rich double-chains were inserted antiparallel in an acidic environment could i-motif be formed [99,100]. Under UV light, MGCB was dissociated into malachite green (MG) cation and OH^-^ ion so that the pH of the solution increased, followed by the i-motif structure being converted into a single chain and the guest molecule being released [98].

For an example of intelligent response to visible light, Yang et al. [94] developed a novel DNA nano-device based on the formation of nano-sized coordination polymers (NCPs) by calcium ion (Ca^2+^) coordination with AS1411 DNA G quadruplex (Figure 7). Since AS1411 was allowed to specifically bind to nucleolin in the nucleus, the delivery system could directly deliver the photosensitizer chlorine e6 (Ce6) to the nucleus to produce ROS in the nucleus. Meanwhile, the delivery system can downregulate the expression of Bcl-2 and trigger the decomposition of endogenous hydrogen peroxide to amend the tumor hypoxia state and enhance the efficacy of PDT. It has been found after administration to tumor-bearing mice that such multiple mechanisms enable the nanocomposites to accomplish high-efficiency synergistic antitumor effects at low doses, with outstanding tumor targeting and light-triggered tumor inhibition.

So far, its application is still limited due to the low tissue transmittance of ultraviolet/visible light and the lack of spatial control of PSs [101]. However, it is gratifying that the near-infrared light has a deep penetration into the tissue, and can be used for irradiation in vivo, which is expected to realize efficient photodynamic therapy (PDT). In 2016, Zhuang et al. [102] devised a triangular DNA nanostructure based on DNA origami for the delivery of a low-molecular carbazole derivative BMEPC in tumor cells. BMEPC is a one-and two-photon imaging agent and photosensitizer that can be fully excited by the near-infrared light to destroy targets under anaerobic conditions. However, the aggregation of BMEPC in the water environment leads to quenching, which limits its application [102]. After binding to the vector DNA containing the largest dense insertion site, the water solubility, photodynamic efficiency, and intracellular photobleaching of BMEPC were significantly improved. The triangular DNA nanostructures loaded with BMEPCs could be successfully ingested by human breast cancer cell MCF-7. Under the irradiation, BMEPCs would generate free radicals and be released due to DNA photo-cleavage, and the generated ROS would eventually lead to cell death. The complex can simultaneously play the roles of an imaging agent and a photosensitizer in cells, and provides new tactics for intracellular imaging and photodynamic therapy.

In addition, the lanthanide-doped upconversion nanomaterials have individual light conversion capability, which can be used to absorb two or more low-energy photons and release a high-energy photon to construct a near-infrared responsive nanoplatform [103]. In recent years, a series of cooperations between the research group of Li Lele, a researcher at the National Center for Nano Science of Chinese Academy of Sciences, and the research group of Zhao Yuliang, an academic at the Chinese Academy of Sciences, has promoted the development of DNA nano medical devices. In the preliminary work, with the purpose of constructing a DNA sensing device that can be activated at a required time under remote stimulation, the Li research group [104] proposed for the first time the remote manipulation of biosensing using up-conversion luminescence. The nano device not only can efficiently deliver the aptamer probe to living cells, but also can time-control the ATP fluorescence sensing activity by in vitro and in vivo near-infrared light irradiation. After that, they expanded and realized the “time-space” controllable and accurate detection and diagnostic applications of a variety of important biomarkers including miRNA and mRNA [105,106,107]. On the basis of these studies, they constructed a DNA nano-device driven by orthogonal near-infrared light [108] (Figure 8a) in this work to achieve higher spatial-temporal accuracy. The nano device combines an ultraviolet light activatable aptamer module and a photosensitizer with up-conversion nanoparticles (UCNPs) [108], and the UCNPs generate orthogonal ultraviolet light and visible light under the excitation of two near-infrared light beams with different wavelengths to activate the biological recognition activity of the aptamer and excite the photosensitizer to generate active oxygen, respectively, thereby realizing space-time controllable target recognition and photodynamic therapy (Figure 8b). Moreover, the DNA nanodevice [108] can be integrated with an immune checkpoint blocking therapy to inhibit the untreated distal tumor by promoting tumor infiltration of cytotoxic T cells and causing synergistic and distal effects in tumor-bearing mice. This work has taken a new step towards the employment of exogenous tools to control the function and behavior of DNA nanodevices in vivo.

Taken together, although ultraviolet light is easily obtained, it will cause damage to biological samples and has insufficient transmission ability, which can be blocked by the skin to reach the lesion site. Nevertheless, near-infrared light can reach a depth of nearly 10 cm in the body and has no damage to normal cells. Therefore, UV is only suitable for in vitro research and is not conducive to clinical application. NIR is a relatively ideal light source for photoresponsive control of drug release due to its strong transmission ability, marginalex absorption by the skin and tissue, and no damage. Under the action of external light irradiation or magnetic field, the inorganic nanomaterials can generate resonance absorption and transfer heat to the surroundings, so that the local temperature increases to cause DNA melting or decomposition, and the drugs stored or blocked by DNA chains are set free. It is possible to design a magnetic response system in addition to the above-mentioned optical response system. For example, Vallet-Regi et al. [109] added superparamagnetic iron oxide nanoparticles to MSN for conjugation with DNA, which enclosed MSN pores when two DNA chains hybridized, and the double-stranded DNA was gradually melted by an external magnetic field (up to a high temperature of 42–47 °C) to result in decapsulation and drug release. This remotely triggered drug delivery is capable of warming that surrounds medium, which may have unexpected effects if applied to the delivery of chemotherapeutic drugs. The principle of temperature-controlled conformational change can also be accessed to design temperature-responsive DNA nanodevices. The Juul team [110] used covalently closed three-dimensional DNA cage structure to encapsulate horseradish peroxidase (HRP) and chose whether to release it under the control of temperature. This structure of 12 double-stranded B-DNA helices forming the edges, each interrupted by a short single-stranded linker forming six “horns” between them [110], allowed the cargo to enter or be released at 37 °C while ensuring that the cargo remained in the central cavity of the cage at 4 °C (Figure 8c). The enzyme trapped within the DNA cage has catalytic activity and the DNA cage device helps reversibly and controllably encapsulate and release the enzyme cargo without any form of covalent or non-covalent attachment to the cargo.

### 3.2. Chemical-Responsive DNA Nanodevices for Drug Carriers

The tumor microenvironment (TME) is featured by being slightly acidic, hypoxic, and rich in tumor cell secretase. Simultaneously, tumor cells also contain high concentrations of reactive oxygen species (ROS), glutathione (GSH), and other biochemical indicators different from those of normal cells [111]. TME-responsive nano-drug carriers need to show outstanding performance in achieving tumor targeting, removal of carrier protective shell, and controlled release of drugs at tumor sites.

pH change is one of the traits of pathological tissues in the human body. Due to the aggressive proliferation of tumor cells and rapid generation of irregular blood vessels, nutrition and oxygen in the tumor site are rapidly lacking, and lactic acid metabolites produced by glycolysis in tumor cells are accumulated in the tumor stroma, resulting in the pH value of the tumor extracellular environment reduced to 6.5–7.2 [112], while the pH values of inclusions and lysosomes in tumor cells are further reduced to 4.0–6.0 [113]. On this basis, a variety of pH-responsive drug delivery systems have been designed to stabilize the vehicle under normal physiological conditions. Ruling out the pH-responsive MUC1-aptamer-DNA icosahedron constructed by Huang’s research group mentioned above, Liu et al. [114] reported a DNA tetrahedral device triggered by an acidic environment, in which the key element was the formation/dissociation of short and cytosine (C)-containing DNA triplexes at pH 5.0–8.0, leading to the assembly or disassembly of DNA tetrahedrons. First, they constructed a three-star motif with three sticky ends through base complementary pairing, and self-assembled into a tetrahedron at pH 5.0. The motif occupied the vertex of the tetrahedron, and the branches of the motif became the sides of the tetrahedron. By dissociating the triad at pH 8.0, the tetrahedron can be restored to its original structure. Therefore, it can also be termed a strategy for reversibly assembling/disassembling DNA nanocages using the DNA nanomachine concept [114]. Recently, Ruan et al. [115] also adopted tetrahedral DNA framework (TDF) as the solid structure framework of the biosensor and DNA i-motif structure consisting of different sequences as the pH response conversion module structure, and developed a programmable pH sensor (Figure 9a). The obvious advantages of this sensor [115] lie in its durative response to biological pH between 5.0 and 7.5, prone to anchoring on the cell membrane and programmability in pH sensing (Figure 9b), demonstrating its potential for dynamically adjustable pH sensing, microenvironment in situ imaging, and pH-related disease diagnosis. In light of the folding synergy, He et al. [116] developed a robust pH sensor with high response sensitivity using dimer DNA i-motif as the module. The DNA framework provided a stable structural framework to manipulate the folding behavior of the responsive nucleic acid module and exert its performance in cells. Dimer DNA i-motif is confined to the vertex of TDF, and the DNA framework promotes the correct assembly of i-motif structure, thus enhancing their folding cooperation; this high-response pH sensor [116] has different sensing ranges from pH 5.0 to 8.0 for monitoring tiny pH changes in cells. The pre-tissue strategy used in this study would be applied to the recent emergence of sensors and molecular switches based on other bimolecular receptors, such as the bimolecular nucleic acid aptamer and the isolated G-quadruplex [117,118,119]. Thereafter, Zhu et al. [120] have devised a DNA nano-hydrogel system for delivering mRNA drugs, which can stably express the target protein in cells. The elaborately designed pH-responsive i-motif “X” DNA scaffold and mRNA were crosslinked to form dense nanospheres for effortless entry into the cells through endocytosis. In lysosomes, with the formation of i-motif, the hydrogel disintegrates and mRNA is released into the cytoplasm encoding protein. Since no other external synthetic materials are used in the system construction process, the system undoubtedly has good biocompatibility, and considering the sophistication of mRNA, the nano hydrogel system is a promising candidate for providing a new way for the delivery of functional mRNA in vivo.

Distinct from the device frameworks in which the abovementioned motifs participate in the construction, the pH-responsive SNA nanodevices based on AuNP designed by Liang team [121] have adjustable pH sensitivity, rapid, and reversible allosteric effects, and even the capability to image lysosomes in living cells. They hybridized a long thioated chain with a short reporter (green) labeled with pH-insensitive Alexa-647 dye, and a tail (red) with a specific protonation site was added at the end of the thioated chain to drive DNA deformation when pH varied [121]. Under the basic condition, the DNA chain was in a linear conformation, and the fluorescence emission of Alexa-647 was quenched. Under the acidic conditions, protonation of the tail would preferentially drive the DNA chain to fold into a loop-stem (green-blue) structure, with the tail region and stem region (blue) forming a three-chain structure through parallel Hoogsteen interactions, with the reporter chain released and followed by generation of measurable fluorescence signals. To confirm its applicability, their team [121] next designed a pH-triggered Dox-SNA conjugate that can transport Dox molecules to cancer cells to promote pH-dependent drug release. The Dox-SNA conjugate can arise by mixing Dox with SNA solution, and thereafter the Dox-SNA compound is internalized by HeLa cells [121]. The results manifested that, also in the acidic environment of late endosome or lysosome, the protonation of the tail (red) forced the chain to adopt the cyclic stem (blue-black) structure, and the Dox loading region (blue) unfolded. When pH was further reduced, the release of Dox was significantly accelerated. In particular, the cumulative release rate of SNA nanocarrier at pH 4.6 reached ~92% within 80 min, illustrating a higher efficiency in drug delivery and cell viability reduction compared with free Dox [121]. Zhu et al. [122] also exhibited a stimulation-responsive multiple drug delivery system capable of accurately responding to pH changes within the range of 5.0–7.0 (Figure 9c) based on a three-strand DNA nano-switch in 2019. Two DNA chains were bound to AuNP. The first DNA chain was an anti-MUC1 aptamer serving as the targeting ligand, and the other was called switchable DNA, which assumed a linear conformation under neutral or basic conditions and self-folded into three chains under acidic conditions. By binding three drugs to this switchable DNA: Dox, cisplatin, and antisense DNA (asDNA), tumor cells undergo a linear to three-chain conformational change of this nanoswitch through endocytosis, leading to the intelligent release of the combined drug. The stimulation-responsive drug delivery system not only has biocompatibility, but also can simultaneously provide multiple therapeutic drugs to improve curative effects. In the tumor-bearing mouse model, we have indicated the effective gene silencing and significant tumor growth inhibition after intravenous injection of the intelligent nano-switch, supplying an opportunity for combined treatment of cancer [122].

In the delivery of tumor vaccine, Ding team [123] attempted to manufacture a tubular DNA nanodevice vaccine capable of delivering antigens and adjuvants, in the hope of specifically stimulating tumor T-cell reaction for a long time to cause tumor regression. Interestingly, the tubular complex was switched by a DNA lock, in a double-stranded state at pH 7.5 and closed along the long sides of the rectangle into a tubular nanostructure with a diameter of about 19 nm, and in a three-stranded state at pH 5.5 resulted in the release of the quencher-labeled pcSW-16 and increased fluorescence intensity. The antigen/adjuvant-loaded DNA nanorobot structure in the lumen can be transported to the draining lymph node (DLN) without the influence of extracellular nucleases. When localized to DLN antigen-presenting cells (APCs), the co-assembled responsive DNA lock is capable of mechanically exposing the drug in response to pH changes, thereby activating the TLR pathway and antigen peptide presentation. The results indicated that the therapeutic vaccination carefully designed by the Ding team could not only affect the primary tumor, but also prevent metastasis and recurrence [123], showing good therapeutic potential and promising to promote the development of personalized cancer vaccine production. The dual-response device engineered by Siwy et al. [124] responds to both pH and electrical signals: lowering the pH to 5.5 will combine positively charged DNA molecules with other chains to form an electrostatic network, thus sealing the pore channels and achieving an unprecedented high resistance of tens of millions of ohms; at a neutral pH value, the voltage switch causes the DNA chain to conduct nano mechanical movement for separation. This pH-dependent reversible closure mechanism proved robust, not only enabling closure and release as a switch, but dominating the drug release rate as a valve.

### 3.3. Biological Responsive DNA Nanodevices for Drug Carriers

Using biological molecules as the stimulation source is a common type of intelligent responsive controlled release, and one of the most commonly used biological molecules is adenosine triphosphate (ATP). ATP is formed by the connection of adenine, ribose, and three phosphate groups. More energy is released during hydrolysis; it is the most direct energy source in organisms, and plays a vital role in major biological processes. Including the common muscle contraction, synthesis and degradation of intracellular compounds, and active transportation, etc., the infinite proliferation of tumor cells demands a large amount of energy to maintain growth and metabolism, resulting in different ATP concentrations inside and outside of the tumor cells. The intracellular ATP concentration is markedly higher than that outside the cell. Studies have found that this property has a palpable effect on the drug resistance of cancer cells. Such systems usually connect the aptamer to the carrier as the control switch: in the absence of ATP, the aptamer or the structure constructed based on the aptamer blocks the drug; in the presence of ATP, the aptamer binds to it closely, leading to the structural transformation of the aptamer and direct realization of drug release [125]. Classical examples include the sandwich-shaped ATP-responsive controlled release system composed of MSN capped with aptamers designed by Tang et al. [126]. In this system, in the absence of ATP, ATP aptamers hybridized with two single chains to form a sandwich DNA structure, which was grafted onto the surface of MSN by click chemistry, accompanied by pore blockage and inhibition of release. In the presence of ATP, ATP binds to the ATP aptamer and leaves the pore canal, and the loaded molecule is released. Subsequent study demonstrated not only high selectivity of this ATP-responsive behavior for ATP analog, but also that the amount of molecular leakage in the absence of ATP could be negligible. Similarly, Gu et al. [127] demonstrated that, when the environment was rich in ATP, the polymer nanocarrier functionalized with DNA motifs bound by ATP aptamers could selectively release the embedded Dox through conformational transformation. On the other hand, the controlled release of ATP-responsive nanogel drug-loading system in tumor-bearing mice was further studied. The system not only effectively inhibited the activity of tumor cells, but also did not affect the body weight and normal cell metabolism of mice [127], which was denoted as a momentous step for its feasibility study in clinical application.

Compared with the controlled release of ATP response, there is also an adenosine response system. Cyclic guanosine monophosphate (cdGMP, second messenger) molecules with intracellular information transmission can also be used for controllable release [128] to integrate cdGMP aptamers into DNA icosahedron. In the presence of cdGMP, the aptamers are changed in structure through chain replacement, thereby destroying the DNA icosahedron structure and opening the capsule to release the drugs therein. Meanwhile, the fluorescence spectrum was used to monitor the release of the goods, further realizing the rigorous control on the opening of the DNA icosahedron and the release of the goods.

As we all know, many reducing molecules (such as glutathione and reductase) in tumor cells generate large redox potential differences inside and outside those cells. Mutual transformation of glutathione (GSH) and glutathione disulfide (GSSG) is a major redox pathway in mammalian cells that is primarily responsible for preventing ROS-induced cell injury, maintaining the stability of thiol-containing enzymes, and preserving the integrity of cell membranes. The rapid proliferation of tumor cells results in the GSH content in tumor tissues being at least four times higher than the normal tissue content, reaching 0.5–10 mM [129]. This significant difference in GSH level can be utilized to trigger the controlled release of redox-sensitive nanodrugs in tumor cells. The routinely used structures for constructing reduction-responsive carriers, such as disulfide bonds (S-S) and selenium bonds (Se-Se), can be rapidly cleaved by high concentrations of GSH. ROS, as a class of molecules with strong oxidizing ability, is a by-product of aerobic metabolism in organisms, and is a special cell signal transcription factor. Hydrogen peroxide (H_2_O_2_), the most typical ROS, has been widely explored as an activation condition for drug release due to its catalytic activity in multiple reactions [130,131]. Compared with normal cells, tumor cells will produce higher concentrations of H_2_O_2_ [132]. The high biotoxicity of H_2_O_2_ is largely due to its easy conversion to various types of active free radicals, which disrupt the redox metabolic balance. In 2018, Ding et al. [44] created a novel drug delivery system based on DNA nanostructure (called TODP), the linear tumor treatment gene p53 was linked using disulfide bonds, and Dox was inserted into the base pair of DNA double chains. Through GSH response, specific release of Dox and p53 in tumor cells was achieved, as well as efficient gene transfer. The system illustrated an effective inhibitory effect on tumor growth in vivo and in vitro, without obvious systemic toxicity. These structural and chemical consensus nanocarriers not only realize the combination treatment of gene and chemotherapy, but offer a new strategy for gene therapy of various diseases. Then, the Ding research group [133] developed a nano-platform based on branched DNA structure to efficiently load the sgRNA/Cas9/antisense complex targeting the tumor-related gene PLK1 through DNA self-assembly for co-delivery of gene editing and gene silencing tools. The gene editing tool sgRNA/Cas9 can be targeted for DNA in the nucleus and the gene silencing tool antisense can be used for mRNA in the cytoplasm, thereby attaining synergistic treatment on tumors in vivo and in vitro. As the structure introduces the aptamer for targeted delivery and the influenza hemagglutinin peptide for endogenous escape, and can effectively release drugs in response to GSH and RNase H, the structure displays high-efficiency antitumor activity without causing visible adverse reactions. This first instance of a nanoplatform based on branched DNA offers new proposals for the development of gene therapy.

Researchers have shown that the pathology of neoplastic diseases is associated with enzyme dysfunction or dysregulation of its expression. Among them, the expression and activity of proteases and phospholipases in tumor tissue are significantly higher than those in normal tissue, which has an important impact on the occurrence and development of tumors [134]. Based on this feature, the design and development of enzyme-responsive antitumor nano-drug delivery systems are of great priority, contributing to the controlled release of drugs at tumor sites [69,135]. Apart from the above-mentioned cases where DNA devices realized the release of drugs in response to enzyme [81,82,133], the conformationally switchable intelligent nano-probe designed by Li research group [136] contains a dense shell of hairpin DNA labeled with gold nanoparticles (AuNP) and 5- carboxyfluorescein (FAM). Under the influence of high-activity telomerase, the conformation of hairpin DNA is transformed, in order to facilitate the drug Dox loaded by AuNP and responding to telomerase activity being released in cancer cells. Analogously, functional DNA tetrahedral probes designed by Yue et al. [137] based on FRET can also respond to changes in telomerase activity, and the probes loaded with telomerase primers and molecular beacons MB can image telomerase in living cells and distinguish between normal cells and tumor cells, with the potential to be utilized for the delivery of tumor drugs. Enzyme molecules may well interact with other responsive factors, and therapeutic effects may be stronger. Zhu et al. [138] then synthesized a biological porous nanosphere (PRS) using RNA as a structural unit and cyclodextrin as a binder. RNA contained EpCAM aptamer for targeted delivery and siRNA (gene drug) for EpCAM gene silencing. Cyclodextrin could load sorafenib (chemotherapy drug) for targeted liver cancer treatment through its hydrophobic cavity. As shown in Figure 10a, Dicer enzymes in liver cancer cells could intelligently respond to and degrade the complex and release drugs for synergistic treatment. The measurements on the cell, subcutaneous and in situ levels of tumor-bearing mice show that the composite can significantly inhibit tumor growth, promote apoptosis (Figure 10b), and is safe and nontoxic for organs to be treated.

Most single-stimulus-responsive nanosystems are susceptible to interference, yet the combination of endogenous and exogenous stimuli often emerges unexpected effects on spatio-temporal control. In addition to the multiple response device cases mentioned above, the matrix metalloproteinase (MMP)-responsive AuNP probe (Figure 10c) designed by Yang et al. [139] can be used for tumor-targeted photothermal therapy and drug delivery. Under the action of the MMP generated by the tumor tissue, the DNA ligated on the surface of the gold nanoparticles can hybridize to enable the complex to be in an aggregation state from dispersion, and under the irradiation of in vitro near infrared light, the material generates local high temperature to induce the release of Dox drug, and the combined treatment of enzymatic reaction and near infrared light is successful. The magnetic DNA nanogel constructed by Yao et al. [140] made use of magnetic nanoparticles as the core, and a DNA nanogel layer is synthesized through rolling cycle amplification for loading anticancer drugs. Under the guidance of an external magnetic field, the magnetic DNA nanogel can achieve high-efficiency targeting to tumor cells. Furthermore, the special DNA nanogel can intelligently respond to various stimulation factors such as temperature, pH, and nuclease, thereby operating the release of anticancer drugs. The system cleverly combines the magnetically controlled delivery with the intelligent response release to further the development of precision medicine. The mutual echo of this exogenous stimulus and endogenous stimulus strikingly enhanced the accuracy of the work of the nano-devices, which might be the development direction of space-time controllable nano-devices in the future.

## 4. Conclusions and Perspectives

Since DNA has been found to have highly programmable and predictable properties, it has been used as a fascinating functional material to design and assemble various nanostructures to exert unique advantages in medical diagnosis and treatment. In addition to being constrained by the principle of base complementary pairing, changes in DNA nanostructure abnormalities under endogenous and exogenous stimuli also suggest the possibility of the development of smart responsive nanodevices. In this review, we initially introduced the concept of DNA nanomaterials and their applications in the medical field, and summarized the classification of DNA nanostructures for drug carriers; especially, the recent accomplishments in the design and application of DNA origami, DNA tetrahedral structure, and spherical nucleic acid were described. Afterwards, we then focused on introducing various intelligent responsive DNA nanodevices for targeted therapy from the physical, chemical, and biological perspectives according to the differences in stimulation sources, confirming that both static and dynamic DNA nanocarriers can be designed to actively and passively release loading substances at specific sites. Collectively, DNA nanostructures, whether as nanocarriers or incorporated in intelligent responsive controlled release systems, have demonstrated good stability and controllability, providing an infinite prospect for their application in drug therapy systems.

However, as a novel thing in the field of drug delivery, DNA nanostructures are facing many challenges and opportunities. Particularly in practical applications, the circulation, distribution, metabolism, efficacy, and degradation of drugs in the body are crucial for studying the interaction between DNA and cell behavior. Although researchers have designed a variety of nanoparticles to transport drugs to solve the problems of toxic and side effects and drug resistance of antitumor drugs, as shown in Figure 11, there are still potential risks [13,49,141]. (1) DNA itself, as a deoxynucleotide polymer, is easily degraded in the blood circulation to reach the target, or generates a certain degree of immunogenicity due to self-modification, thereby interfering with the function of the immune system of the body. How to reasonably design DNA nanostructures to avoid the special hazards of nanomaterials needs to be explained in more detailed research. (2) Due to the small size effect and biocompatibility of DNA nanomaterials, after crossing the cell membrane, some regions of DNA devices may bind to single-stranded RNA such as mRNA and genes in the nucleus to regulate gene expression, resulting in gene expression disorder and protein dysfunction. Therefore, the timely clearance mechanism of DNA nanomaterials plays an irreplaceable role. (3) Some DNA nanomaterials need light, magnetic or other external assistance to be effective, and the safety effect on normal cells around the tumor remains to be proved. Therefore, DNA sensors that require the application of external conditions often need to be verified multiple times to ensure that toxic and side effects are minimized.

Undoubtedly, benefiting from the biocompatibility and degradability of DNA nanomaterials, in the future, functional DNA will be developed into more complicated and advanced forms to achieve more efficient medical applications, such as multi-responsive delivery of nano-robots. However, because of the late research in this field, most of the studies so far have been limited to simply revealing the operations at the laboratory level, and no related drugs have been marketed. Therefore, considering the actual starting point, the next stage of technical research should also include standardization converted to the clinical level. Additionally, the high-purity preparation and large-scale production of DNA nanostructures are also great challenges impacting their commercial applications, which are necessary conditions for practical biomedical applications. Last but not least, the price of DNA nanostructure synthesis has also received a high degree of attention. Complex nanocarrier synthesis means that more DNA chains are desired, so reducing the cost of synthesizing nanostructures has become the focus of research. It is believed that, with the rapid development of intelligent DNA nanocarriers, customized personalized DNA nanocarriers will provide a powerful security means for the effective treatment of tumors, and have more extensive applications in the field of precision biomedicine.

At present, most of the studies are limited to the level of cells, tissues, or tumor-bearing mice. To extend this drug delivery system to human trials, there are still many problems to be solved, such as population control, energy supply, and material selection. We anticipate that, in the next two decades, it will develop into a higher clinical stage, even in small-scale disease applications. However, FDA approval could be more difficult because the range of interests affected by this technological revolution cannot be underestimated. Scientists still have a long way to go.

## Figures and Tables

**Figure 1 biomolecules-11-01855-f001:**
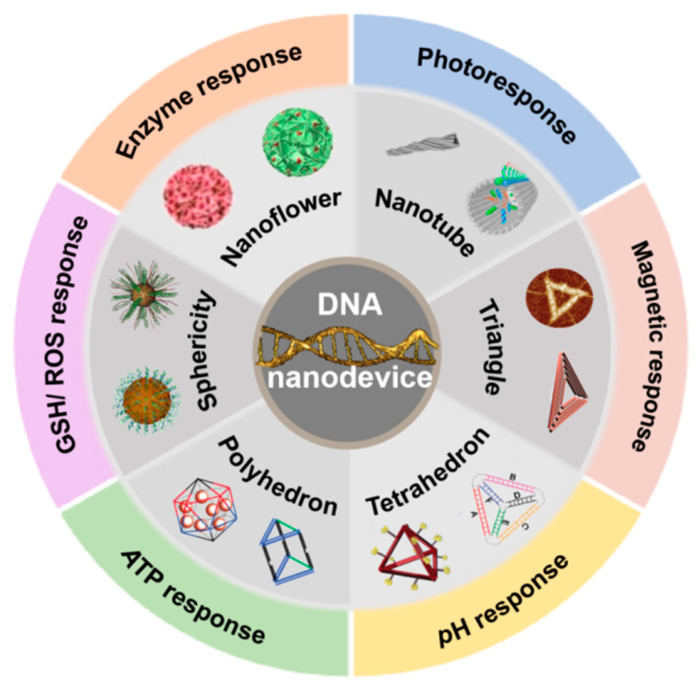
Schematic illustration of different types associated with DNA nanodevices.

**Figure 2 biomolecules-11-01855-f002:**
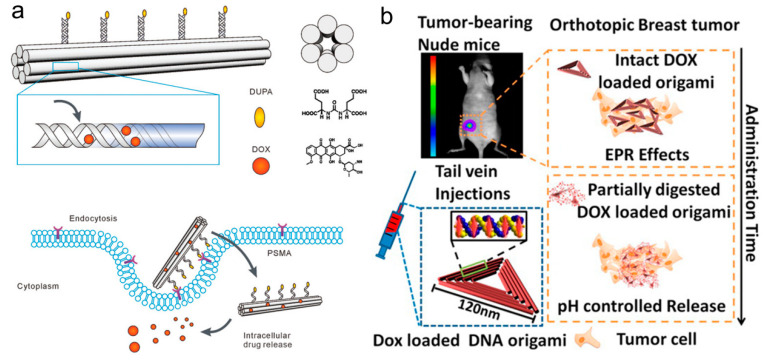
Role of nanostructures constructed based on DNA origami in tumor targeted therapy. (**a**) ADC-like nanocomposites: principles of construction and process of operation, reproduced with permission from Reference [42], Copyright 2020 John Wiley and Sons; (**b**) The Dox/DNA origami complexes injected into the tail are transported by blood circulation and accumulate in nude mouse mammary tumors due to the EPR effect, reproduced with permission from Reference [53], Copyright 2014 American Chemical Society.

**Figure 3 biomolecules-11-01855-f003:**
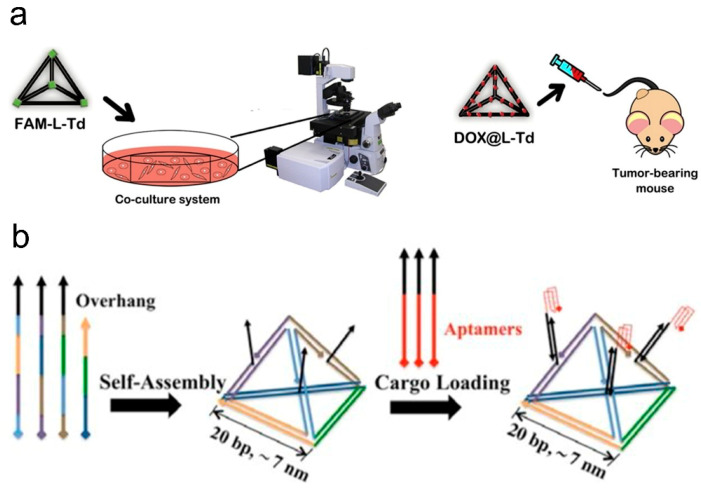
Structural assembly and functional characterization of tetrahedral DNA. (**a**) Laser scanning confocal microscopy (LSCM) was used to observe the uptake of L-DNA tetrahedrons (L-TDs) of different sizes by living cells over time and to test the potential of L-TDs to deliver low-dose Dox in mice, reproduced with permission from Reference [59], Copyright 2017 Elsevier. (**b**) AS1411 aptamers are attached to multiple overhangs on the tetrahedral edge of DNA to target nucleolin receptors on the plasma membrane of cancer cells, reproduced with permission from Reference [67], Copyright 2014 American Chemical Society.

**Figure 4 biomolecules-11-01855-f004:**
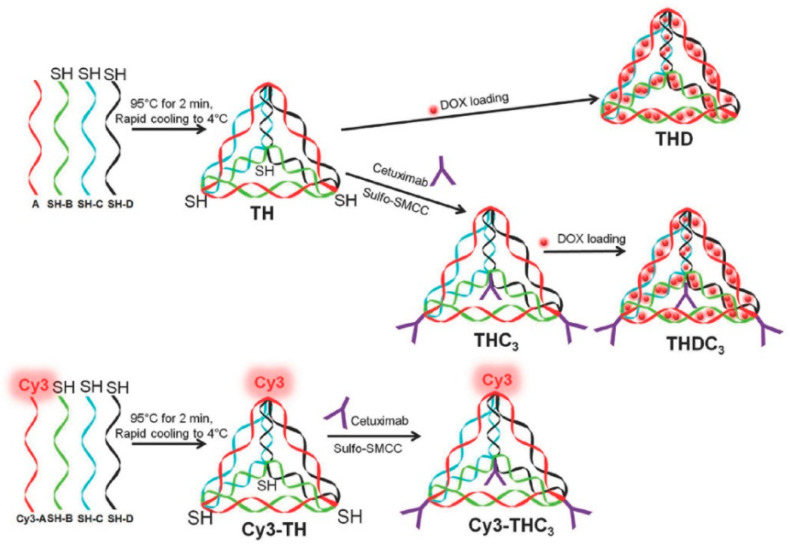
Structure formation diagram of DNA tetrahedral nanocomposites. DNA TH was self-assembled by four ssDNAs in the previously reported way and could be directly loaded with Dox to form THD, or coupled to cetuximab (anti-EGFR antibody) via the sulfo-SMCC to form THC3, which could then be loaded with Dox to form THDC_3_. When the A chain is replaced by the Cy3-A chain, Cy3-TH and Cy3-THC_3_ are generated in steps so that these structures can be detected, reproduced with permission from Reference [72], Copyright 2016 John Wiley and Sons.

**Figure 5 biomolecules-11-01855-f005:**
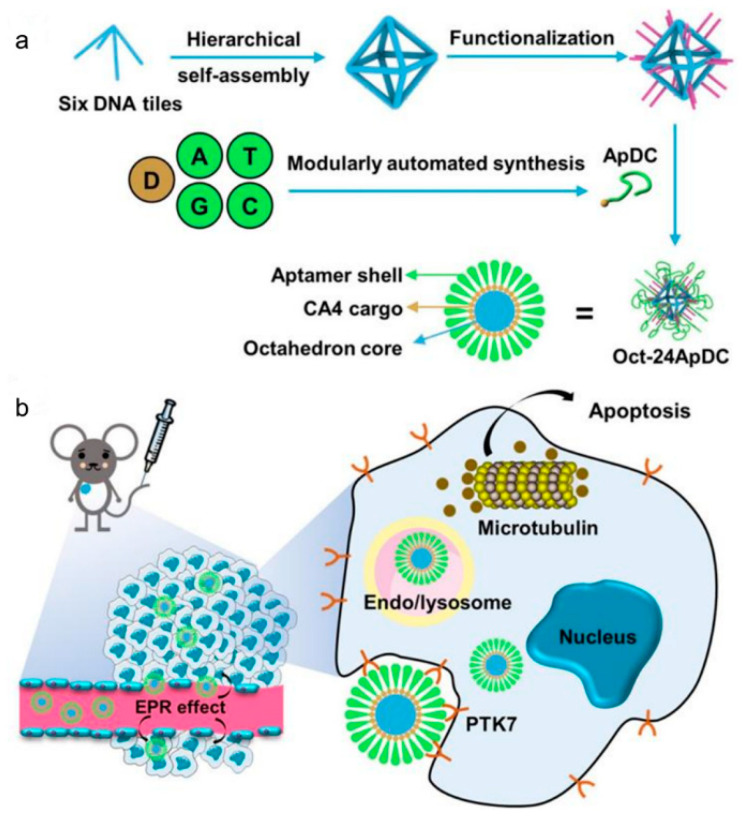
Schematic illustration of targeted delivery of CA4-FS by DNA octahedron wireframe. (**a**) Six DNA tiles are layered and self-assembled to form an octahedral framework; after the nucleic acid aptamer is functionalized, single-chain handles connected on each side of the framework can be combined with a synthesized CA4-FS module; and with designing a sequence of the single-chain handles, different numbers of CA4-FS can be connected. The DNA nanocarrier in core-shell mode is finally formed, which is called CA4-Oct, reproduced with permission from Reference [79], Copyright 2020 American Chemical Society; (**b**) structural advantages: The outer three-dimensional spatial distribution of multivalent Sgc8c aptamers allows more precise binding to the tumor marker PTK7 receptor; enhanced permeability and retention (EPR) of passive targets improves delivery efficiency, reducing direct exposure of CA4 to avoid premature biotoxicity; the dense DNA octahedral framework facilitates the insertion of flexible CA4-FS into solid tumors to release the drug, reproduced with permission from Reference [79], Copyright 2020 American Chemical Society.

**Figure 6 biomolecules-11-01855-f006:**
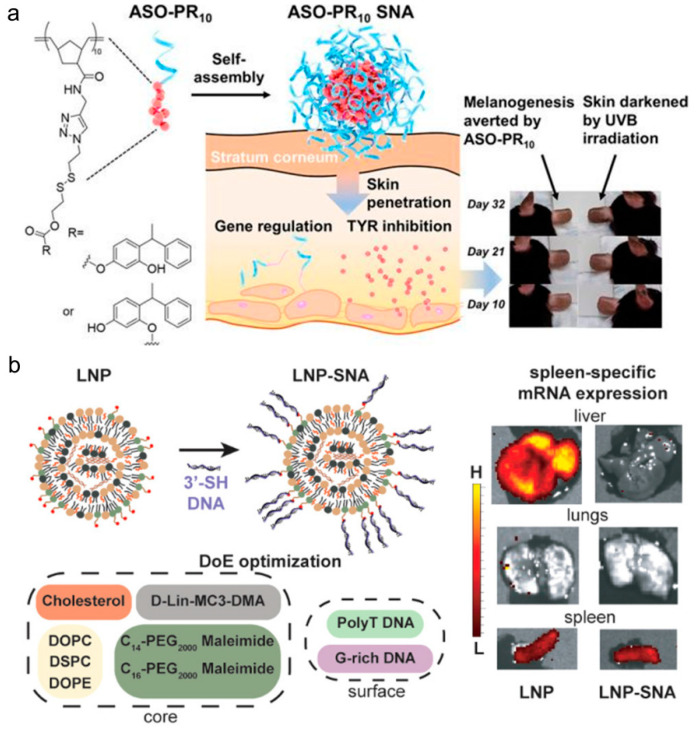
Different spherical nucleic acid (SNA) structures for drug targeted delivery. (**a**) SNA-based ASO-drug conjugates exert anti-melanoma effects as the skin depigmentation agent, reproduced with permission from Reference [88], Copyright 2021 American Chemical Society; (**b**) optimization of the composition of the lipid nanoparticle SNA (LNP-SNA) and changes in the biodistribution and therapeutic properties of the structure in mice compared to conventional LNP, reproduced with permission from Reference [90], Copyright 2021 American Chemical Society.

**Figure 7 biomolecules-11-01855-f007:**
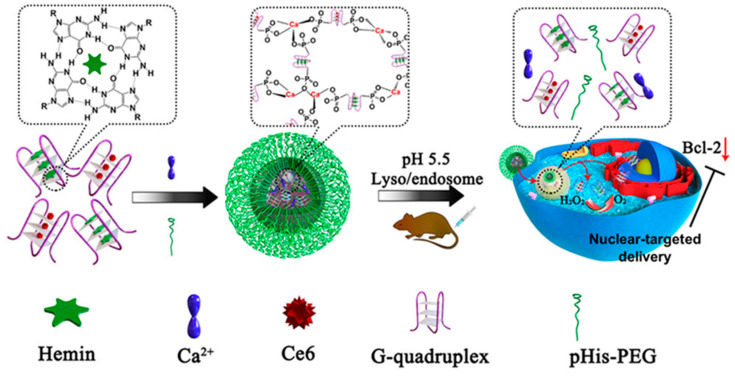
Schematic diagram of intranuclear delivery of photosensitizer using nanoscale coordination polymers (NCPs) prepared based on DNA, reproduced with permission from Reference [94], Copyright 2018 American Chemical Society.

**Figure 8 biomolecules-11-01855-f008:**
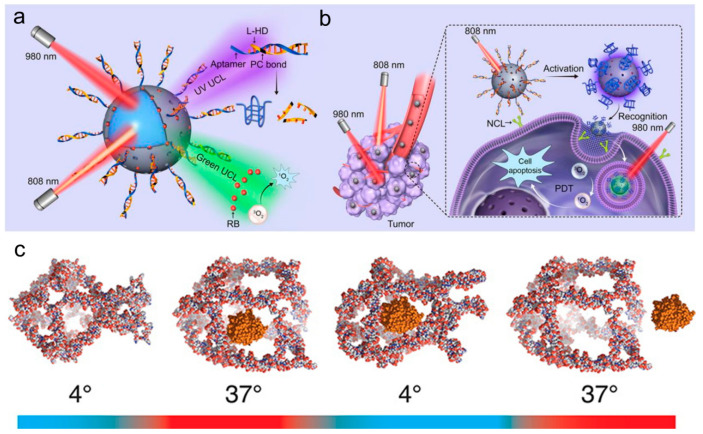
Photothermal-responsive DNA nanodevices for drug carrier. (**a**) UCNPs (light modulator) are used to convert NIRs of two different wavelengths (808 and 980 nm) into orthogonal ultraviolet light and up-converted light (UCL), respectively, and (**b**) the nanodevice recognizes nucleolin on the surface of tumor cells under local trigger of 808 nm NIR light and then generates ROS under irradiation of 980 nm NIR light for PDT, reproduced with permission from Reference [108], Copyright 2020 The Authors; (**c**) the encapsulation and release of HRP by a covalently closed three-dimensional DNA cage device are controlled by temperature, reproduced with permission from Reference [108], Copyright 2013 American Chemical Society.

**Figure 9 biomolecules-11-01855-f009:**
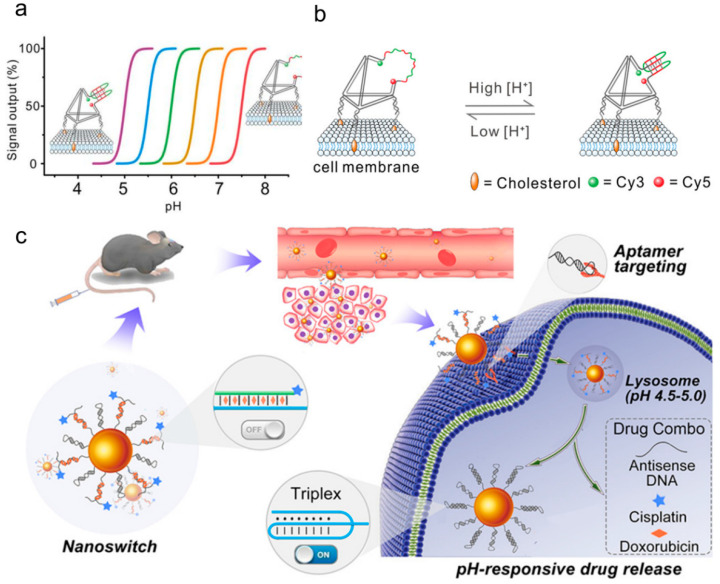
pH-responsive DNA nanodevices for drug carrier. (**a**) The TDF-based programmable pH sensor using DNA i-motif as the proton recognition probe provides a continuously distributed pH response over the range of 5.0 to 7.5. and (**b**) the i-motif-TDF nanosensor was anchored to the cell surface for in situ pH analysis, reproduced with permission from Reference [115], Copyright 2021 American Chemical Society. (**c**) The triplex-DNA nanoswitch-based system responds accurately to pH changes over the 5.0–7.0 range. In an extracellular neutral space, a linear DNA nanoswitch is used for fixing a plurality of therapeutic drugs; After entering the intracellular acidic space, the DNA nanodevice changed from linear to triple to release the drug, reproduced with permission from Reference [122], Copyright 2019 American Chemical Society.

**Figure 10 biomolecules-11-01855-f010:**
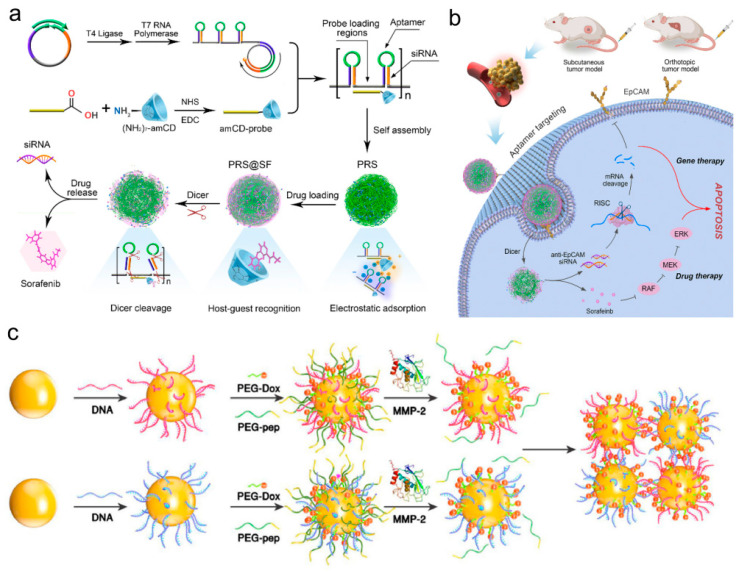
Enzyme-responsive DNA nanodevices for drug carrier. (**a**) The enzymatic RNA generated by rolling circle transcription (RCT) was self-assembled into PRS by using cyclodextrin as a complexing agent, and SF was further carefully loaded to form porous RNA nanospheres (PRS@SF), and (**b**) after intravenous injection of PRS@SF, the PRS@SF was transported in blood vessels and swallowed by target cells, and the drug SF and siRNA were released for synergistic treatment after being digested by cytoplasmic Dicer enzyme, reproduced with permission from Reference [138], Copyright 2020 Elsevier. (**c**) MMPs induced specific and rapid assembly of PEG-Dox-attached AuNPs in vivo to enhance tumor inhibition, reproduced with permission from Reference [139], Copyright 2019 Elsevier.

**Figure 11 biomolecules-11-01855-f011:**
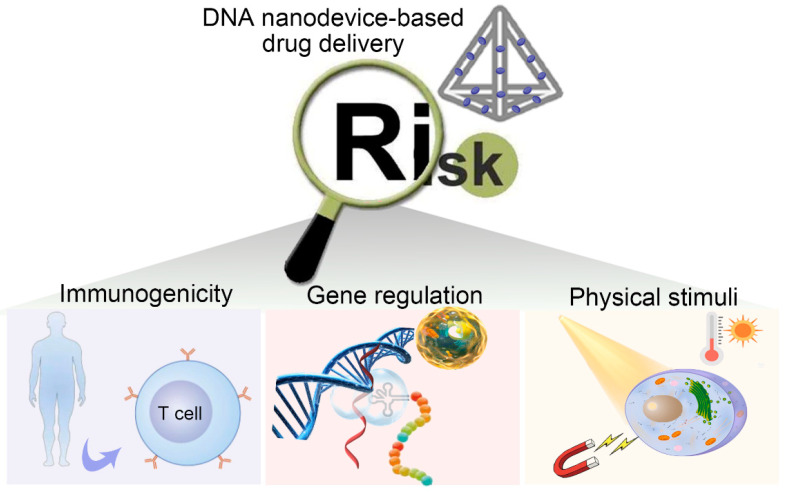
Potential risks of DNA nanotechnology for drug delivery systems.

## Data Availability

Not applicable.

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
