# Peer review of "DNA Nanodevice-Based Drug Delivery Systems"

_biomolecules, 2021, doi:10.3390/biom11121855_

Round 1
Reviewer 1 Report
Certainly this is interesting review on the application of DNA structures for drug delivery. In my opinion, more Figures should be added for a better understanding of such complex molecules. Regarding the language, I would recommend a deep revision. Some points to be clarify or corrected:
Abstract: L 15 “stimuli.”
L18. Please, specify the period of time
L28. 28 ?
L588, “people”?
L628 “silicon”?
L757 pH5.0”
L801 “Dox” or DOX ?
L854 “nano device for drug carrier”
L894 “masses of reducing molecule”
Author Response
Reviewer #1:
Certainly this is interesting review on the application of DNA structures for drug delivery.
Response: Many thanks for the valuable comments.
- In my opinion, more Figures should be added for a better understanding of such complex molecules.
Response: Thank you for your valuable comment. As the reviewer suggested, we have added three images and the corresponding description in the hope of elucidating the DNA nanoparticles as clearly as possible in the revised manuscript, figure 3, figure 7 and figure 11, respectively (Page 7, Line 266-274; Page 16, Line 667-670; Page 29, Line 1039-1040).
- Regarding the language, I would recommend a deep revision. Some points to be clarify or corrected:
Abstract: L15 “stimuli.”
L18. Please, specify the period of time
L28. 28 ?
L588, “people”?
L628 “silicon”?
L757 pH5.0”
L801 “Dox” or DOX ?
L854 “nano device for drug carrier”
L894 “masses of reducing molecule”
Response: Many thanks for the helpful comments. In addition to the problems listed, we have checked the entire review to avoid recurring errors of the same type. Corresponding corrections have been supplied in the revised manuscript. (Page 1, Line 15, 18, 27-28; Page 3, Line 115, 117; Page 4, Line 137, 143, 173, 175-176, 178; Page 5, Line 209; Page 6, Line 231, 233, 257; Page 8, Line 311, 313, 333; Page 9, Line 365, 371, 373; Page 10, Line 378; Page 12, Line 494, 502; Page 13, Line 521, 532; Page 15, Line 599, 637; Page 16, Line 657; Page 17, Line 703, 710; Page 18, Line 739, 744, 771; Page 19, Line 779, 781, 816, 825; Page 20, Line 829; Page 21, Line 861, 868; Page 22, Line 908, 927-928; Page 23, Line 966, 970, 974; Page 24, Line 993).
Reviewer 2 Report
The review article “DNA nanodevice-based drug delivery systems” by Chaoyang Guan et al. aims to document the progress in the field of DNA based nano devices with vast biomedical applications. It is timely review considering the demand for novel delivery systems at the time of ongoing pandemic. The authors have reviewed vast majority of the literature and clearly presented it along with helpful illustrations.
I am enthusiastic about this review article and supportive of its publication. I only offer some minor suggestions to improve readability and enhance the message of the paper (adopting them is optional).
Minor issues:
- Authors have addressed the positive side of the DNA based nano drug delivery systems. But there is need to address the potential risks and side effects of such technology. One illustration showing all reported risks will add value to the review.
- Authors should also consider adding one paragraph on the regulatory framework surrounding these DNA nanodevice-based delivery systems. How close are we to the FDA approvals?
Author Response
Reviewer #2:
The review article “DNA nanodevice-based drug delivery systems” by Chaoyang Guan et al. aims to document the progress in the field of DNA based nano devices with vast biomedical applications. It is timely review considering the demand for novel delivery systems at the time of ongoing pandemic. The authors have reviewed vast majority of the literature and clearly presented it along with helpful illustrations. I am enthusiastic about this review article and supportive of its publication. I only offer some minor suggestions to improve readability and enhance the message of the paper (adopting them is optional).
Response: Many thanks for the valuable comments.
- Authors have addressed the positive side of the DNA based nano drug delivery systems. But there is need to address the potential risks and side effects of such technology. One illustration showing all reported risks will add value to the review.
Response: Thanks a lot for the helpful comment. As the reviewer suggested, we have added a summary of the risks of DNA nanotechnology with a corresponding illustration in the revised manuscript (Page 24, Line 1018-1019; Page 25, Line 1020-1040, Page 30, Line 1344-1345).
- Authors should also consider adding one paragraph on the regulatory framework surrounding these DNA nanodevice-based delivery systems. How close are we to the FDA approvals?
Response: Thank you for your valuable and thoughtful comment. It is really a good suggestion for adding one paragraph on the regulatory framework surrounding these DNA nanodevice-based delivery systems. As is suggested, we have added a new paragraph at the end of the article. Corresponding descriptions have been supplied in the revised manuscript (Page 26, Line 1057-1063).